# Methodology and Neuromarkers for Cetaceans’ Brains

**DOI:** 10.3390/vetsci9020038

**Published:** 2022-01-21

**Authors:** Simona Sacchini, Pedro Herráez, Manuel Arbelo, Antonio Espinosa de los Monteros, Eva Sierra, Miguel Rivero, Cristiano Bombardi, Antonio Fernández

**Affiliations:** 1Veterinary Histology and Pathology, Veterinary School, Institute of Animal Health, University of Las Palmas de Gran Canaria, c/Transmontaña s/n, 35416 Arucas, Spain; pedro.herraez@ulpgc.es (P.H.); manuel.arbelo@ulpgc.es (M.A.); antonio.espinosa@ulpgc.es (A.E.d.l.M.); eva.sierra@ulpgc.es (E.S.); miguel.rivero@ulpgc.es (M.R.); antonio.fernandez@ulpgc.es (A.F.); 2Department of Veterinary Medical Science, University of Bologna, Ozzano dell’Emilia, 40064 Bologna, Italy; cristiano.bombardi@unibo.it

**Keywords:** cetaceans, dolphins, beaked whales, neuroanatomy, neuropathology, methodology, immunohistochemistry, neuromarkers

## Abstract

Cetacean brain sampling may be an arduous task due to the difficulty of collecting and histologically preparing such rare and large specimens. Thus, one of the main challenges of working with cetaceans’ brains is to establish a valid methodology for an optimal manipulation and fixation of the brain tissue, which allows the samples to be viable for neuroanatomical and neuropathological studies. With this in view, we validated a methodology in order to preserve the quality of such large brains (neuroanatomy/neuropathology) and at the same time to obtain fresh brain samples for toxicological, virological, and microbiological analysis (neuropathology). A fixation protocol adapted to brains, of equal or even three times the size of human brains, was studied and tested. Finally, we investigated the usefulness of a panel of 20 antibodies (neuromarkers) associated with the normal structure and function of the brain, pathogens, age-related, and/or functional variations. The sampling protocol and some of the 20 neuromarkers have been thought to explore neurodegenerative diseases in these long-lived animals. To conclude, many of the typical measures used to evaluate neuropathological changes do not tell us if meaningful cellular changes have occurred. Having a wide panel of antibodies and histochemical techniques available allows for delving into the specific behavior of the neuronal population of the brain nuclei and to get a “fingerprint” of their real status.

## 1. Introduction

In the course of evolution, cetaceans have undergone modifications with respect to their ancestral terrestrial status. Throughout the phylogenesis, the cetaceans’ sphlancnocraniums lengthened in order to obtain a greater hydrodynamic body shape, the upper airways (the blowhole) migrated dorsally, while the neurocranium shortened (Figure 1). The skull acquired a post-orbital compression and an antero-orbital elongation, called “cranial telescoping” [1]. However, the brain does not diminish its size; it folds over itself and thus acquires a “boxing glove” shape, also due to the lack of most of the olfactory structures in the frontal lobes and the pronounced width of the temporal lobe (Figure 2, right). One of the most eminent transformations occurred in the toothed whales (odontocetes) in their size, structure, and neuroanatomical organization. A unique characteristic of the toothed whales is the exceptionally large size of the brain, in some species both in absolute and in relative terms [2], and the extremely dense folding of the neocortex [3] (Figure 2, right). The size of the dolphin’s brain compared to body size varies between apes and humans [4]. Toothed whale brains may range from about 200–2000 g [5] to 9300 and 9200 g, the maximum sizes found in the killer whale (*Orcinus orca*) [6] and the sperm whale (*Physeter macrocephalus*) [7], respectively. In addition, female sperm and killer whales evolved a brain larger than expected for their body mass (encephalization quotient > 1), with values near the ranges of primates, dolphins, and elephants [8].

The animal cerebral tissue is fragile due to the general lack of connective tissue and the high percentage of lipid content and the post mortem period negatively affects the conservation, quality, and stiffness of brain tissue. Undisputedly, where the best possible morphology is required, animals should be anesthetized and subjected to cardiac perfusion with saline, followed by a formalin flush [9,10]. In any event, the study of cetaceans provides valuable information on the conditions of our seas. Cetaceans are homeotherms, long-lived species, and are located at the top of the marine food chain. Thus, they are considered bioindicator species and sentinels of the health of the sea. The lifespan of these animals can be compared, in many species, to the one of humans, and makes cetaceans, particularly odontocetes, a new and more authentic comparative natural model for the study of certain neurodegenerative diseases in humans [11,12,13,14,15]. Although immersion fixation has been considered inadequate, especially in large brains, where the fixative is slow to penetrate [16] and different protocols have been proposed for cetaceans’ large brains [2,17], it is possible to obtain viable material for the microscopic study of cortical [18,19,20,21], subcortical [20,22,23], and brainstem regions [14,20,24,25]. With this in view, we validated a methodology in order to preserve the quality of such large brains (neuroanatomy/neuropathology) and at the same time to obtain fresh brain samples for toxicological, virological, and microbiological analysis (neuropathology). Moreover, a fixation protocol adapted to brains, of equal or even three times the size of human brains, was studied and tested.

## 2. Materials and Methods

### 2.1. The Workflow for Cetaceans’ Brain Examination: From Fresh to Fixed Samples

From 2009 to 2016, 281 cetaceans stranded and died in the Canary Islands (Spain). Systematic pathological studies were performed on the carcasses to determine the cause of death and/or stranding. Brains were obtained from 51 specimens of six different species of the suborder Odontoceti: striped dolphin (*Stenella coeruleoalba*) (*n* = 10), Atlantic spotted dolphin (*Stenella frontalis*) (*n* = 7), common dolphin (*Delphinus delphis*) (*n* = 3), bottlenose dolphin (*Tursiops truncatus*) (*n* = 3); short-finned pilot whale (*Globicephala macrorhynchus*) (*n* = 3), Risso’s dolphin (*Grampus griseus*) (*n* = 5), pygmy sperm whale (*Kogia breviceps*) (*n* = 2), Blainville’s beaked whale (*Mesoplodon densirostris*) (*n* = 5), Cuvier’s beaked whale (*Ziphius cavirostris*) (*n* = 8), Gervais’ beaked whale (*Mesoplodon europaeus*) (*n* = 4), and True’s beaked whale (*Mesoplodon mirus*) (*n* = 1). The bottlenose dolphins, one newborn and two adults, came from a controlled environment and died of natural causes. The postmortem times were different, due to the intrinsic logistic aspects of each stranding, but never exceeded 48 h (decomposition code 1 and 2, very fresh and fresh).

### 2.2. Opening of the Skull

The necropsy protocol was carried out as standardized and published by Thijs Kuiken and Manuel García Hartmann in 1991 [26], with some modifications and added contributions. The opening of the cranial cavity was performed with a swinging saw. Four lines were drawn in the skull and a window was created in order to offer the best access to the brain, as follows:one dorsal parallel to the nuchal ridge (crista occipitalis externa) at 1 cm from it (slightly more caudally in some species such as beaked whales or BW and pilot whales),another ventral and parallel to (a), bypassing the occipital condyles; andtwo lateral and perpendicular to (a) and (b), passing through the parietal and the squamosal bones, in the temporal fossa.

In the biggest size animals (i.e., sperm-, baleen-, and beaked- whales), the aforesaid window should be drawn more caudally. In fact, the thickness of the skull, and sometimes the ossification of the falx cerebri and tentoria cerebelli [27], cripples any effort to achieve the brain and force to draw the window just around the occipital condyles.

### 2.3. Cautious Sampling of Fresh Brain at Necropsy: A Key Step

Brains were removed, coded for freshness, and usually dissected within 24–48 h after death. A superficial sampling of fresh unfixed brain was usually done, in order to complement investigations with microbiological, virological, and toxicological (included biotoxins) studies. The following samples were taken: the cerebral cortex (rostrally and caudally), the pons, the cerebellum, and the medulla oblongata. Then, in order to expose the lateral ventricles, a cut was made in each hemisphere (as shown in the inset of Figure 2).

At this time, a biopsy punch was inserted in the lateral ventricles, first rostral-ward, and then caudal-ward, ensuring in this way to take samples of the caudate nucleus (rostral) and the thalamus (caudal). A sample of the choroid plexus was also taken.

### 2.4. Fixation of Cetaceans’ Brains: A Challenge

Cetaceans’ brains are usually obtained after different post-mortem periods, making it difficult to control autolysis times. The post-mortem times are different but should never exceed 24 h, in particular for those brains used for neuroanatomical studies. Brains were immersion-fixed at the time of necropsy in 10% neutral-buffered formalin (4% formaldehyde solution, pH 7.4).

Due to the great encephalic volume and the slow rate of diffusion of the formalin (0.5 cm per hour), some longitudinal cuts (2 to 4) were made in both cerebral and cerebellar hemispheres, before immersion. Cuts were mainly superficial (Figure 2, right) but in each cerebral hemisphere, at least one of them entered deep in order to expose the lateral ventricles and to allow the fixative to go inside the ventricular system (the same cut as Section 2.2). The opening of the ventricular system has been proposed by other authors [9,10] and has been adapted here to the characteristic short ventricular system of these marine mammals [28]. To respect the optimum volume ratio of fixative tissue, which is 10–20:1 [9], brains were dipped in a container with a minimum volume of 20 L of fixative. Absorbent paper was placed at the bottom of the container to avoid sticking of the brain to the plastic surface. Brains were left in the fixative for at least 72 h, and the correct fixation was monitored throughout the days.

### 2.5. Sectioning of the Whole Brain and Postfixation

Brains can then be sectioned in sagittal, transverse (cross) or dorsal sections. However, cross sections were usually chosen because they offer a better evaluation of neuropathological lesions (i.e., uni- or bilateral distribution of the lesions, affected areas, etc.) and a finer study of the neuroanatomy [9]. A dual sided edged knife for precision brain sectioning or a slicing machine were used for cross-cutting. Sections (1–1.5 cm thick) were returned to the fixative for at least other 48 h before proceeding to the final sampling of the brain. Indeed, sections were then stored in a 4 or 10-L volume container with fresh fixative, separated from each other by filter paper.

### 2.6. Routinary Neuropathologic Investigations (FFPE)

For neuropathological purposes, we used the routine Formalin-Fixed Paraffin-Embedded (FFPE) procedure. Regrettably, unlike domestic mammals, our greatest guide for sampling is directed only by gross findings, since we do not usually have any information on neurological or diagnostic examination.

A representative sampling of the main areas of the brain has been made as follows (Figure 3):-telencephalon: cortex (2 to 4 samples, at least frontopolar and occipital cortex) (a and f, stars), corpus striatum (b, arrowheads), amygdala (c, blue circle; h, Am), and hippocampus (h, hip);-diencephalon: thalamus and hypothalamus (c, star and rectangle);-mesencephalon: tectum (colliculus rostralis and caudalis) (d, star) and tegmentum (with the substantia nigra) (d and i, arrowheads);-rhombencephalon: at least pons (with the locus ceruleus) (e, rectangle), trapezoid body (with the cochlear nuclei) (f and k, rectangle), medulla oblongata (g, rectangle) and at least two samples of the cerebellum (included a sagittal section of the vermis) (e and g, arrowhead and star);-choroid plexus (d, g, i, j, and k, circle);-spinal cord (Figure 2, left): at least pars cervicalis and pars thoracica.

Hippocampus is a complex brain structure embedded deep into the temporal lobe. In these animals, it may be a hard issue to detect it, due to its particular small size [23]. However, it can easily be found out just behind the large amygdaloid complex (Figure 3h, hip).

The selected samples were then placed into standard tissue cassettes measuring 30 × 25 × 4 mm and postfixed for at least other 24 h. Larger size processing cassettes (Super Mega-Cassette System, size 70 × 50 × 15 mm) could be a better option. These types of cassettes allow the processing and/or embedding of large sections of brain but processing schedules must be adjusted accordingly.

Fixed brain samples were dehydrated through a series of graded ethanol baths and then infiltrated with paraffin wax. Tissues were embedded in paraffin wax blocks and processed for routinary haematoxylin eosin staining and, when necessary, histochemical procedures were used to identify microorganisms, intra/extracellular material (i.e., Periodic acid–Schiff, PAS) as well as to distinguish neurons, glial cells, neurofibrils, and myelin (i.e., Nissl, Bielschowsky, Luxol Fast Blue, etc.).

### 2.7. Immunoperoxidase Staining: Paraffin Embedded Tissues (p-IHC)

Immunohistochemistry has been established as a reliable methodology for both routine diagnostic and research activities in veterinary medicine. Paraffin embedded samples were deparaffinated with xylene and rehydrated in graded ethanol. Immunohistochemistry was carried out with standard immunohistochemical Avidin-Biotin Complex (ABC) protocol [30]. Endogenous peroxidase activity was blocked by incubation with 3% H_2_O_2_ in methanol for 30 min at room temperature (RT). Antigen retrieval was performed depending on the primary antibody (Table 1). Primary antibodies were selected after reading up in the literature. Some of them were already used in other animal species (i.e., tyrosine hydroxylase or calbindin) [31,32] while others were picked up from human investigations (i.e., α-synuclein or β-amyloid) [33,34]. While polyclonal antibodies usually recognize many epitopes and give immunohistochemical positive results, we also obtained promising results employing monoclonal antibodies as well (Table 1). Any biotin blocking reagent was used to reduce the background from endogenous biotin. Sections were incubated overnight at +4 °C with the primary antibody. Negative controls, consisting of the omission of the primary antibody and incubation with only 10% normal serum in phosphate-buffered saline (PBS), were performed for all the immunohistochemical assays.

The day after, sections were rinsed in PBS (three times, 5 min each) and incubated for 90 min at RT with a secondary antibody (diluted 1:200) in a solution containing 10% normal serum in PBS (Table 1). Sections were rinsed in PBS and incubated with an avidin-biotin complex (ABC, Vector Laboratories; PK-4000) for 1 h at RT and developed using the 3,3′-diaminobenzidine (DAB) peroxidase kit (Vector Laboratories, SK-4100) or the 3-amino-9-ethylcarbazole (AEC) peroxidase kit (Vector Laboratories, SK-4200). DAB sections were then dehydrated in ethanol, cleared in xylene and coverslipped with DPX (Sigma). In addition, after rinsing, some DAB immunolabeled sections were counterstained using thionine prior to dehydration and coverslipping. AEC sections were counterstained with a non-commercial Mayer’s hematoxylin and coverslipped with an aqueous-based mounting medium (Vector).

### 2.8. Cryoprotection and Preparation of the Sample (FFCS)

For neuroanatomical purposes, a selective sampling of the brain was made. Cryoprotection, as well as cryosectioning, ff-IHC, and Nissl staining (see next sections), were adapted to previous published protocols for rat brains [37,38]. It is important to leave the samples for a further period of 48 h in 10% neutral-buffered formalin or 4% paraformaldehyde (preferably under agitation). In this way, we can ensure an optimal fixation and stiffness of the samples. Later, in order to remove the excess of fixative, the samples were washed for 48 h in PBS, under agitation and changing the buffer at least 2–4 times. After rinsing in PBS, the samples should be cryoprotected by immersing them in a 30% sucrose and 0.1% sodium azide solution in PBS (pH 7.4) at +4° C, in order to avoid freezing artifacts [39]. The solution has the properties of protecting and nourishing the piece of tissue (sucrose) by preserving it in a buffered liquid (PBS), free of bacteria and fungi (sodium azide). When the sample gets soaked with the solution, it sinks to the bottom of the container, and it is considered fully cryoprotected and ready for the next step. When using a cryostat, samples were immersed in a mixture of PBS-sucrose and Optimum Cutting Temperature formulation (OCT) (1:1) overnight. The day after, the samples were included in a mold using the same mixture and were rapidly frozen, and 40–50 μm-thick serial sections were obtained with a cryostat [(Leica CM1950, Nussloch, Germany), University of Las Palmas de Gran Canaria]. Sections were stored in PBS (pH 7.4) solution with sodium azide (0.01%). On the other hand, when using a freezing sliding microtome [(Leica SM 2000 R), University of Bologna], samples were directed removed from the cryoprotecting solution, fixed to the sample holder stage, and rapidly frozen. Working temperatures ranged between −20 °C and −35 °C, depending on the external temperature. The sucrose crystallizes creating the sample’s adherence to the stage. The samples were then sectioned at 50–60 μm.

### 2.9. Immunoperoxidase Staining: Free-Floating Immunohistochemistry (ff-IHC)

Sections proceeding from formalin fixed free-floating sections were treated with 3% H_2_O_2_ in PBS for 30 min at RT, in order to eliminate endogenous peroxidase activity, and rinsed in PBS (three times, 10 min each). To block non-specific binding, sections were incubated in a solution containing 10% normal serum, and 0.5% Triton X-100 (Merck, Darmstadt) to permeabilize the tissue in PBS for 2 h at RT. Thereafter, a first set of sections were incubated in the primary antibodies (Table 1), during 18 h, at +4 °C. Negative controls consisting of the omission of the primary antibody and incubation with only 10% normal serum in PBS were performed for all the immunohistochemistry experiments. After 18 h, sections were rinsed in PBS (three times, 10 min each) and incubated for 45 min at RT with a secondary antibody (diluted 1:200) in a solution containing 1% normal serum in PBS (Table 1). Some primary antibodies may require a 36 h period of incubation, in refrigeration at + 4 °C. Keeping the sections in immersion permits their conservation and prevents them from drying out. The sections were successively rinsed in PBS and incubated with an avidin-biotin complex (ABC, Vector Laboratories; PK-4000) for 1 h at RT. Finally, the sections were developed using a 3,3′-diaminobenzidine (DAB) peroxidase kit (Vector Laboratories, SK-4100) and mounted on coated slides to dry overnight. DAB Slides were then dehydrated in ethanol, cleared in xylene and coverslipped with DPX (Sigma). AEC sections were coverslipped with an aqueous-based mounting medium (Vector). In addition, some DAB immunolabeled sections were counterstained using thionine (Section 2.10) prior to dehydration and coverslipping.

### 2.10. Nissl Staining

For Nissl staining, thionine (Lauth’s violet) was the election dye. Thionine is a strongly metachromatic dye, useful for the staining of acid mucopolysaccharides. It is a common nuclear stain and can be used for the demonstration of Nissl substance in neurons. Sections were taken out of the first 24-well plate containing PBS with sodium azide, mounted on gelatin-coated slides, and dried. The sections were defatted and soaked for 1 h in a mixture of chloroform and 100% ethanol (1:1), rehydrated through a graded series of 100%, 96%, 80%, 70%, and 50% ethanol and distilled water (2 min each), stained 30 min in a 0.125% thionine (Fisher Scientific, Waltham, MA, USA) solution, rapidly dehydrated (few dips each step), left 2 min in 100% ethanol, cleared 10 min in xylene, and coverslipped with Entellan (Merck, Darmstaldt, Germany).

### 2.11. TUNEL Staining

In order to provide in situ detection of apoptosis in brain tissues, using a TUNEL (TdT-mediated dUTP-biotin Nick End-Labeling) based assay, the two following kits were used: TACS^®^ 2 TdT-DAB In Situ Apoptosis Detection Kit Reagent (Trevigen, 4810-30-K) and NeuroTACS™ II In Situ Apoptosis Detection Kit (Trevigen, 4823-30-K). The TACS^®^ 2 TdT-DAB Kits utilize a cation optimization system to enhance the in situ detection of apoptosis in a TUNEL based assay. Enzymatic incorporation of biotinylated nucleotides in fragmented DNA is performed by the TdT enzyme. Biotin labeling is detected using streptavidin-horseradish peroxidase and DAB. The manufacturer step-by-step protocol was used, with the only exception that samples were then counterstained with thionine instead of methyl green, due to the better results given by thionine. Two positive controls were used in each experiment: a sample of a cetacean prescapular lymphatic ganglion and a brain sample incubated with TACS-Nuclease.

An OLYMPUS BX41 light microscope was used for the histological and histopathological observations with the support of the Digital software Imaging Solutions, CellA.

## 3. Results

### 3.1. Evaluation of Brain Quality

In order to achieve an optimal fixation of the brain, it was fundamental to:-Provide longitudinal cuts to expose the lateral ventricles and allow the entry of the fixative;-Make cross-sections of the brain after at least 72 h of immersion in the fixative. Once a great percentage of the fixation process was achieved, serial cuts of the brain allowed a greater fixation;-Finally, the post-fixation of the selected samples permitted to complete the fixation of the tissues and provide the necessary firmness for their next processing;-Reposition of new fixative was made after cross sectioning the brain (immersion in a smaller container during 48 h) and then after sampling (postfixed during 24 h).

Sections proceeding from brains, which have not been processed as indicated in the protocol of manipulation and fixation, generated drawbacks and artifacts like:-Tissue rupture and wearing, dark, shrunken and pycnotic neurons of that brains incorrectly handled during necropsy;-Poor fixation of the deepest subcortical structures of those brains, which did not receive cross-sections. This resulted in poor tissue quality, predisposing to easy rupture of the sections, especially during the continuous manipulation in ff-IHC, poor immunoreactivity to the neuromarkers, and a very low affinity to thionine;-The prolonged permanence in the fixative resulted in the loss of tissue quality, which predisposed it to an easy rupture of the sections, loss of antigenicity, and a very low affinity to thionine. In addition, formalin pigment accumulation was observed, as a background deposition and occasionally within the neurons mimicking the neuromelanin pigment;

Finally, frozen unfixed and uncryoprotected brains were shrunken, very breakable during ff-IHC, and presenting vacuolations.

### 3.2. Neuromarkers for Neuroanatomical Studies

Histochemical thionine Nissl staining has been a key dye to explore most of the brain areas and nuclei of the examined species. Nissl staining offered a first evaluation of the brain quality. Alternatively to thionine, cresyl violet may be used; however, thionine provided better coloration of neurons in cetacean brain samples. Some of the investigated brain areas were the red nucleus and the nucleus ellipticus, the motor nucleus of the oculomotor (III) nerve, the motor nucleus of the trigeminal (V) nerve, the cochlear nuclei (the nucleus cochlearis ventralis, VCN, and the nucleus cochlearis dorsalis, DCN; Figure 4a), the motor nucleus of the facial nerve, the olivary nuclei, the hippocampus (Figure 4b), the claustrum, the amygdaloid complex (corpus amygdaloideum), the corpus striatum, the hypothalamic nuclei (Figure 4c), the locus ceruleus, the substantia nigra (Figure 4d), as well as other areas of the prosencephalon and the rhombencephalon. VCN and DCN, whose locations vary between the medulla oblongata and the pons according to mammalian species, have been localized near the trapezoid body in all the examined cetacean species.

In addition, neuromarkers as Tyrosine Hydroxylase (TH), Vasopressin, Corticotropin Releasing Factor (CRF), Glial Fibrillary Acidic Protein (GFAP), Calretinin, Calbindin, and Parvalbumin (Table 1) have been used to identify and deeply characterize different structures of the toothed whale brain. Key areas of the stress circuitry have been investigated: the amygdaloid complex (calbindin), the paraventricular, supraoptic (Figure 4c), and suprachiasmatic (Figure 4c) nuclei of the hypothalamus (vasopressin and CRF), the locus ceruleus and the substantia nigra (Figure 4d) (TH). TH-immunoreactive neurons have also been identified in the hypothalamus. In addition to thionine, calbindin D-28k allowed a complete description of the 12 deep, superficial, and other nuclei which make up the amygdaloid complex. Special interest has been given to the central nucleus of the amygdala, a key protagonist of the stress system (unpublished data). The use of the immunohistochemical technique against the antibody calbindin D-28k permitted a better definition and identification of the limits of the central nucleus of the amygdala. This nucleus was identified as a triangular area, presenting an intense immunoreactivity in the neuropil. Cortical neurons have been explored through parvalbumin, calbindin, calretinin, and nNOS (Figure 4e,f). Finally, GFAP has been used to detect normal, reactive, or neoplastic astrocytes.

### 3.3. Neuromarkers for Neuropathological Studies

Some neuromarkers for infectious (Distemper Virus, Herpesvirus type 1), neoplasic (GFAP), and neurodegenerative diseases (neurofibrillary tangles, β -amyloid, α-synuclein, ubiquitin, and laforin) have also been tested (Table 1). Vascular Aβ deposits (Figure 4i) as well as intranuclear Aβ immunoreactivity and amyloid plaques were also found. NFT are composed of insoluble paired helical filaments of a highly phosphorylated form of the microtubule-associated protein τ (tau) and associated lipids. NFT positivity has been observed both as uniform cytoplasmic or as granular to vacuolar cytoplasmic inclusions (Figure 4h). Alpha-synuclein was detected as round PAS-negative bodies both intraneuronal and/or in the neuropil (Figure 4g). At the same time, these structures presented positivity to ubiquitin antibody (Figure 4g, inset).

### 3.4. Neuromarkers for Acoustic Trauma Research

A mild nuclear staining with TUNEL assay was detected in the VCN, as in the case of the Cuvier’s BW shown in Figure 4j, mainly in the multipolar small spherical and multipolar globular neurons of the VCN. Immunohistochemistry revealed different patterns of expression of c-fos, c-jun, HSP70, ubiquitin, neuroglobin, and nNOS in the cochlear nuclei of the toothed whales. Well preserved brain tissues usually presented a clear c-fos and c-jun nuclear staining. Ubiquitin has been detected both intranuclear (Figure 4k) and in the cytoplasm. HSP70 immunostaining was observed both intranuclear and as cytoplasmic granular deposits. nNOS and neuroglobulin immunopositivity was localized as uniform cytoplasmic staining (not shown).

## 4. Discussion

One of the main challenges of working with cetaceans’ brains is to establish a valid methodology for an optimal manipulation and fixation of such specimens, for neuroanatomical and neuropathological studies. Such difficulties are related to (1) their large brain, (2) the logistic difficulties and laboratorial complexity to achieve a large sample size from certain elusive species, such as the BW, and (3) the difficulties to obtain extremely fresh brain samples from stranded individuals [15].

Veterinary pathologists face many challenges when performing immunohistochemistry because of the diversity of the studied species, which does not guarantee antibodies to cross-react among different species [30]. This is particularly true for pathologists working with cetaceans’ tissues. The postmortem period and the conservation procedures directly influence the good results of the immunohistochemistry. In the same animal, it is possible to observe very well-preserved superficial brain areas and, at the same time, poorly stained unpreserved deep areas.

A representative sampling of the main areas of the brain is always the best option both in human [40] and veterinary [41] neuropathology, in particular when any macroscopic lesion is present. However, there is a fine equilibrium between fresh brain sampling and the preservation of the neuroanatomy. Reading up in the literature, it turns out that there is a great heterogeneity in cetacean brain protocols, mainly due to the diverse purposes of the studies (neuroanatomy vs. neuropathology), as well as in the use of different fixatives [2,14,17,24].

In the present study, we described the methodology for large brain sampling, sectioning, and staining. A panel of 20 useful antibodies has been tested; some of them have been previously used in different published works or are part of unpublished manuscripts. When establishing and planning the cetacean brain sampling protocol, it is crucial to consider some important aspects which have been uncovered in our study:-Several infectious pathogens including virus, bacteria, fungi, and parasites might cross the blood-cerebrospinal fluid barrier, entering the central nervous system and leading to inflammatory infectious diseases like meningitis and meningoencephalitis [42], very common in these animals;-Opening the ventricular system is a crucial step which allows its checking for exudates, space-occupying lesions, asymmetries, abnormal cerebrospinal fluid, and/or any alterations affecting the choroid plexus (i.e., cystic lesions as in Figure 3j or swelling). It is highly important to sample the choroid plexus, as a fundamental site of invasion of bacteria, virus (distemper), and protozoa. In fact, lesions may only be confined to the periventricular areas [43];-Thus, the opening of the ventricular system ensures a rapid uniform penetration of the fixative and the best possible preservation of the tissues. It is important to respect the proportions of fixative because of the large size and rounded shape of cetaceans’ brains. As fixative molecules bind to the tissue, they are depleted. Inadequate fixative volume will result in inadequate tissue fixation [44];-No macroscopic changes may be detected during necropsy and sampling but severe histopathological hallmarks may be present;-Random slicing and sampling of the brain may result in confused neuropathological interpretations. In addition, a strong knowledge of neuroanatomical structures is a critical advantage in order to boost the interpretation of neuropathological changes and their etiopathogenesis. Hence, another important aspect is respecting the international anatomical terminology (*Nomina Anatomica*) of the International Committee on Veterinary Gross Anatomical Nomenclature [45], which evolves over time;-A strong sampling protocol should not forbear to preserve the bilaterality of the brain, which permits to draw the specific pattern of distributions of the lesions, the first important step in neuropathological diagnosis [41];-Even if we usually lack clinical data on stranded animals, when brain lesions are the cause of the stranding, they are usually severe enough, large, and/or diffused.

At the same time, an ideal protocol should evolve over time and evaluate the best options as in the case of the fixative. As an example, zinc-ethanol-formalin fixative (ZEF) has been proposed as an excellent (preserved antigens and minimal background staining) and a safer choice for human brain [46]. Histologically, the preservation of cytomorphological features of neurons, neuroglia, and the various neuroparenchymal components accounted for cellular integrity in the tissues evaluated in this study.

This work showed that a brain sampling protocol can easily combine neuroanatomical and neuropathological purposes. In fact, the neuroanatomical areas related to stress circuitries (the amygdaloid complex and its central nucleus of the amygdala, the locus ceruleus, the paraventricular and supraoptic nuclei of the hypothalamus) can be easily identified in the cetacean’s brain using Nissl staining and immunohistochemical procedures [14,20]. Other target neuroanatomical areas (i.e., the cochlear nuclei, the substantia nigra or the hippocampus) have also been identified.

Some neuromarkers for infectious [36,47] neoplasic [48] and neurodegenerative diseases [15] have been assessed. About the last ones, recently, different investigations have shown that cetaceans are prone to suffer from neurodegenerative disease, as Alzheimer’s Disease (AD). Indeed, it has been suggested that dolphins might be one of the very few potential natural models of AD [11,12,13,14,15,49,50,51,52]. Even if it is still not possible to affirm that cetaceans—namely toothed whales—undergo AD, it is certain that neurodegenerative hallmarks clearly come forth in the brain of the toothed whales. When planning a sampling protocol, including areas like the amygdaloid complex, the hippocampus, the locus ceruleus, and the substantia nigra enables veterinarians to investigate neurodegenerative diseases thanks to an associated panel of neuromarkers.

In the same way, α-synuclein, ubiquitin, and laforin immunopositivity have been checked summing three important neuromarkers for the brain differentiation of globular bodies (namely, Lafora bodies, Lewy bodies-LB-, and pale bodies-PB-). LB, a defining pathological characteristic of two important neurodegenerative diseases, Parkinson’s disease (PD) and dementia with Lewy bodies, constitutes the second most common nerve cell pathology, after the NFT in AD [53,54]. Alpha-synuclein has been identified as the main constituent of the LB pathology [55]. Additionally, ubiquitin is expressed by many PB and outer LB rings, being the staining intensity of the former weak, whereas the latter are often strongly stained [56]. LB are also distinguished for being PAS-negative. We have described round α-synuclein and ubiquitin positive, PAS-negative round bodies, in the mesencephalic neuromelanin rich neurons, which might be identified as LB (unpublished data). Finally, we have tested for the first time in the cetacean brain two different α-synuclein antibodies, Syn 505 (Invitrogen) being a better solution and giving better results than the Abcam clone. These types of neurodegenerative inclusions may be often underdiagnosed due to the presence of other round bodies like mucocytes, Buscaino bodies, or corpora amylacea. Even more importantly, the study of these neuronal inclusions will certainly yield more revealing findings and etiopathogenetic elements on neurodegenerative hallmarks in toothed whales.

Neuronal (nuclear), neuropil (amyloid plaques), and vascular (deposits) Aβ-immunolabeling have been observed in some animals included in the present study. Aβ-immunolabeling was also investigated in one of the 14 BW stranded in close temporal and geographic association with an international naval exercise (Neo-Tapon) held on 24 September 2002 [57,58]. In this BW, we observed intranuclear cortical neuronal Aβ expression suggesting a possible neuroprotective role to hypoxia [15]. Interestingly, in the same animal, apoptotic neurons were observed in the VCN. In addition, immunohistochemistry revealed different patterns of expression of c-fos, c-jun, HSP70, ubiquitin, neuroglobin, and nNOS in the VCN of the toothed whales, including the BW. Specific attention has been given to BW, especially those who died in close temporal and geographic association with an international naval exercise (unpublished data). Apoptotic neurodegeneration is one example of an abnormality that can be triggered even by very transient toxic exposures [59]. This mechanism has recently been demonstrated in the cetacean neurons. A study conducted on neurons chemically reprogrammed from fibroblasts of mass stranded melon-headed whales (*Peponocephala electra*) and used for in vitro neurotoxicity assays revealed that exposure to 4-hydroxy-2′,3,5,5′-tetrachlorobiphenyl, a metabolite of PCBs, may induce neurodegeneration through disrupted apoptotic processes [60]. In the same way, ubiquitin is produced in response to cell injury, binding proteins toward catabolism. In response to apoptotic stimuli, the ubiquitin-ligase activity of inhibitors of apoptosis can lead to their auto-ubiquitylation and degradation, which allows cells to commit to apoptosis [61]. The study of apoptotic mechanisms may reveal new insights in the cetacean brain. On the other side, c-fos and c-jun have been explored here in the toothed whales for the first time. The proto-oncogene c-fos, an immediate early gene, is expressed in neurons in response to various stimuli [62]. Immunohistochemical c-fos detection is useful for the detection and mapping of groups of neurons that display changes in their activity. Proto-oncogenes c-fos and c-jun are also rapidly activated in the brain following a transient ischemia [63]. Strong expression of the c-jun gene and protein is known to precede or coincide with periods of intense cell death in other different conditions like AD [64] and PD [65]. Anyway, in our study, well-preserved brain tissues usually presented a clear c-fos and c-jun nuclear staining.

Lastly, neuroglobin is involved in enhancing primary hypoxic tolerance in the diving brain [66], scavenging reactive oxygen and nitrogen groups and thus defending against cellular damage [67]. Previous studies have shown that resident neural globin proteins (neuroglobin and cytoglobin) may be inversely related to maximum dive duration in marine mammals. In fact, deep divers seem to use circulating globins in the brain while faster swimming coastal species use enhanced neuroglobin and cytoglobin stores, maybe to defeat the hematocrit shortfall [68].

As previously stated, no macroscopic changes may be detected during necropsy and sampling, but severe histopathological hallmarks may be present. At the same time, histopathological findings may suggest a list of differential diagnoses, but only the use of IHC and other laboratory techniques may allow for determining the specific infectious etiology [29]. Going one step further, many of the classical measures used to evaluate neuropathological changes do not tell us if meaningful cellular changes have occurred [69] or may reveal just a tip of the iceberg. Through the support of modern advances in molecular biology and immunolabeling for assessing protein location and function, through a wide panel of antibodies and histochemical techniques available (included the TUNEL assay), it is possible to delve into the specific behavior of each neuron and to get a “fingerprint” of its real status (Figure 5).

## 5. Conclusions

We thus propose a cetacean brain sampling protocol for neuroanatomical and neuropathological studies, including neurodegeneration. The protocol is complemented by different neuromarkers useful for a complete diagnosis of the brain status. Future studies could fruitfully explore this issue further by checking new neuromarkers in the cetacean brain.

## Figures and Tables

**Figure 1 vetsci-09-00038-f001:**
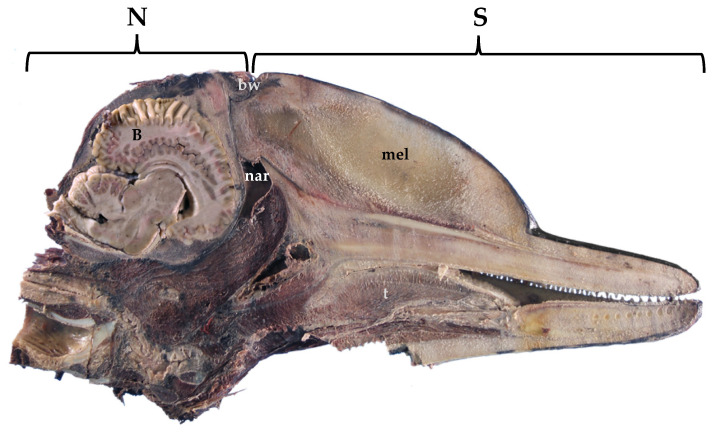
Sagittal section of a fixed dolphin head, showing the lengthening of the sphlancnocranium (S), the shortening of the neurocranium (N), and the position of the brain (B). bw, blowhole; mel, melon; nar, nares; t, tongue. Atlantic spotted dolphin. *Stenella frontalis*.

**Figure 2 vetsci-09-00038-f002:**
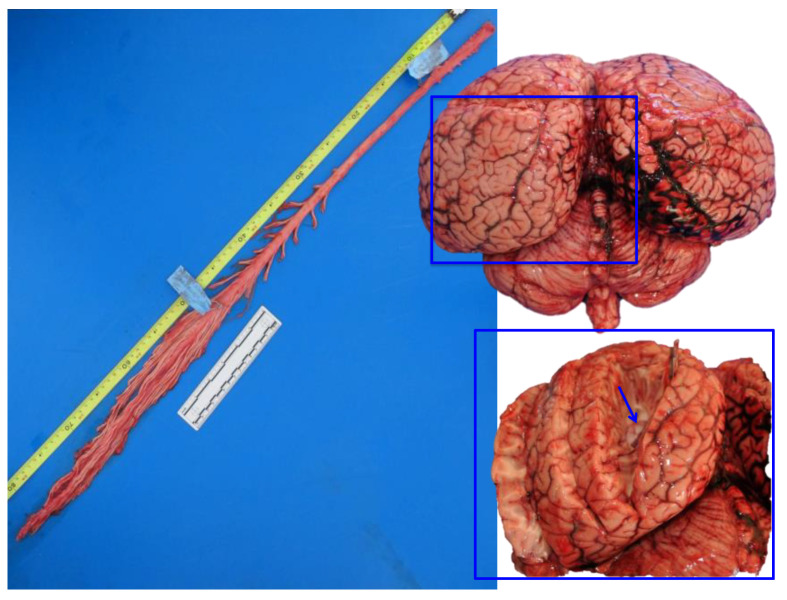
Fresh spinal cord (**left**) and fresh brain (**right**). In the inset, the lateral ventricle is exposed (arrow), in the proximity of the sagittal cleft. Striped dolphin, *Stenella coeruleoalba*.

**Figure 3 vetsci-09-00038-f003:**
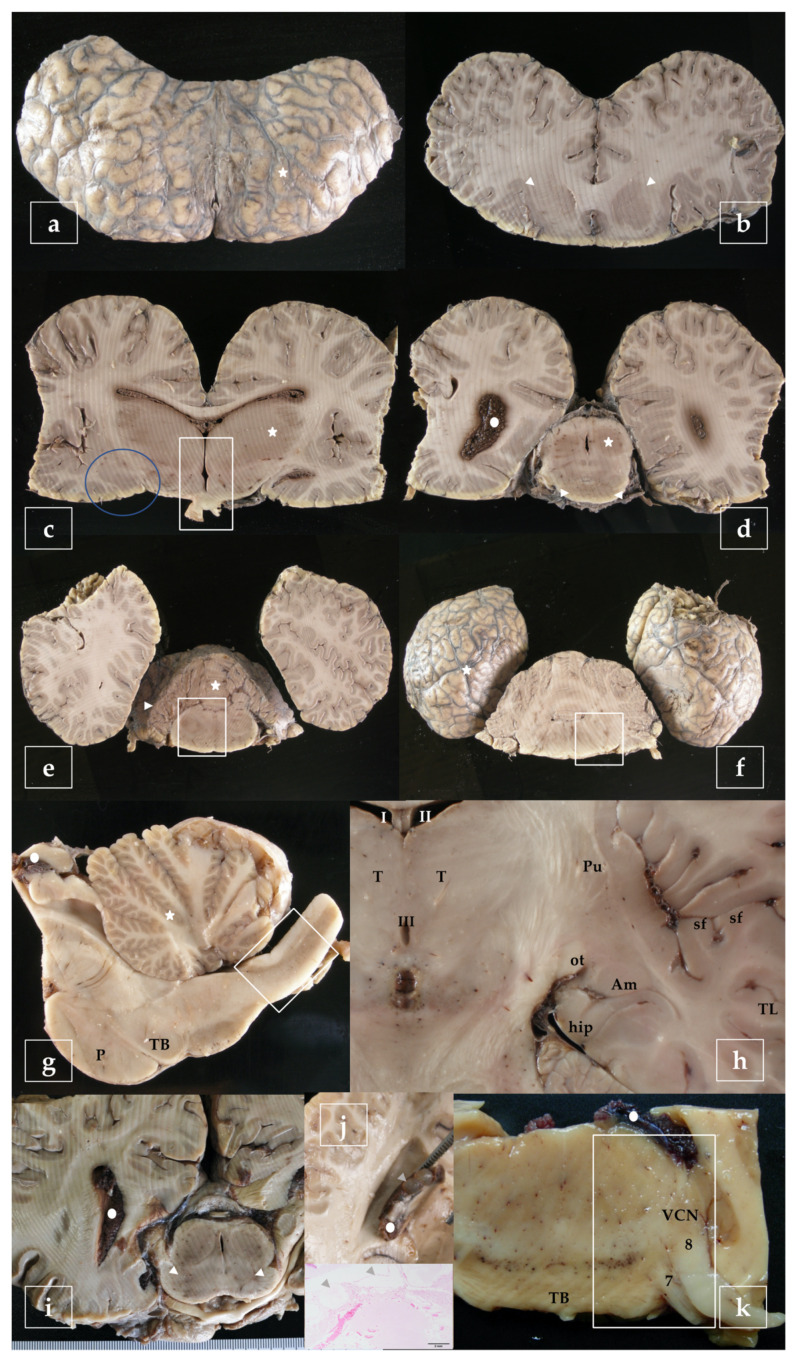
Representative brain sampling in cross sections. Details are given in the main text (Section 2.6). Striations are due to the dentated blade of the slicing machine (i.e., in **b**). Blainville’s beaked whale, *Mesoplodon densirostris* (**a**–**f**). Sagittal section of the brainstem. P, pons; TB, trapezoid body. Atlantic spotted dolphin, *Stenella frontalis* (**g**). Dorsal section of the brain, obtained at the level of the ventral limit of the telencephalon. The amygdaloid complex (Am), is located rostral to the tiny hippocampus (hip) and lateral to the optic tract (ot). Pu, putamen; sf, sylvian fissure; T, thalamus; TL, temporal lobe; I-II-III, 1st-2nd-3rd ventricle. Atlantic spotted dolphin, *Stenella frontalis* (**h**). Cross section made as in (**d**) intended to show the substantia nigra (arrowheads). Gervais’ beaked whale, *Mesoplodon europaeus* (**i**). Cross section of the brain exposing the lateral ventricle. The choroid plexus presents cystic lesions (arrowheads). The animal was positive to herpesvirus and presented lesions in the brain [29]. Striped dolphin, *Stenella coeruleoalba* (**j**). Sample of the brainstem including the trapezoid body (TB), exposing de ventral cochlear nucleus (VCN), the facial nerve (7), and the vestibulocochlear nerve (8). Risso’s dolphin, *Grampus griseus* (**k**).

**Figure 4 vetsci-09-00038-f004:**
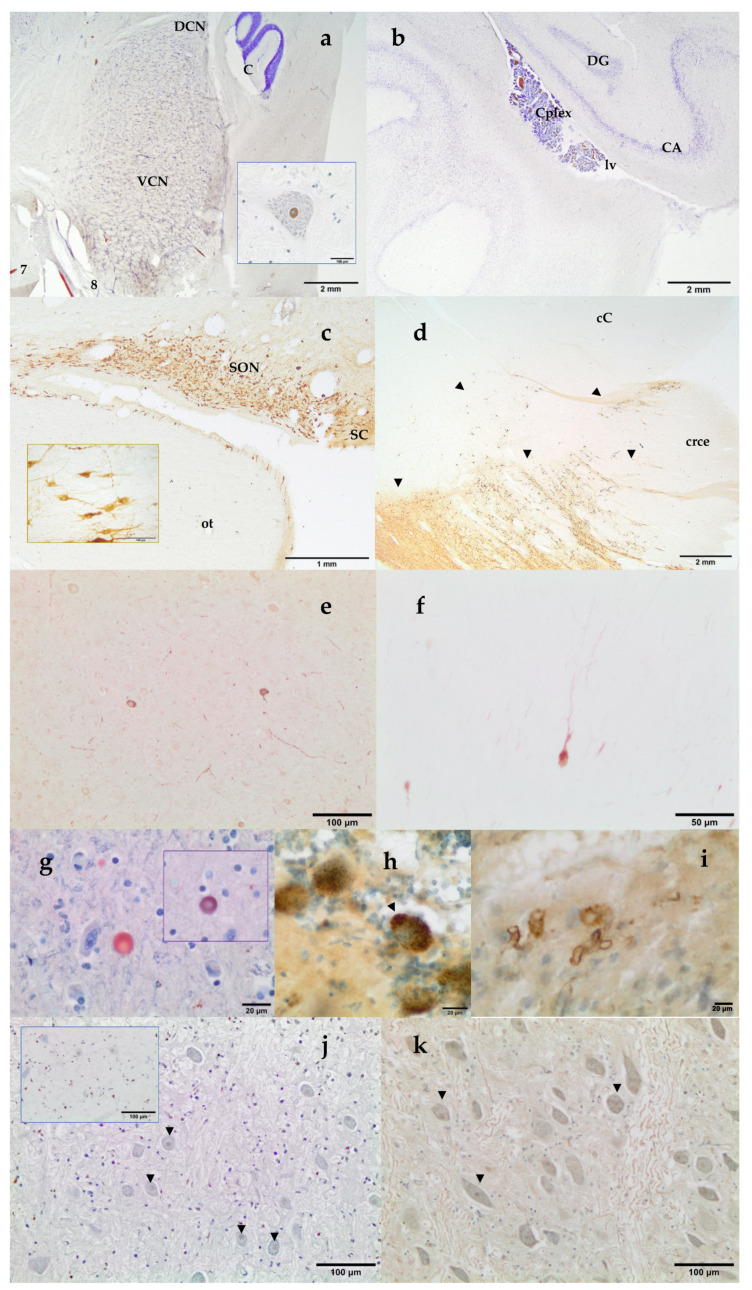
VCN and Dorsal Cochlear Nucleus (DCN). 7, facial nerve; 8 vestibulocochlear nerve; C, cerebellum. Thionine. Striped dolphin, *Stenella coeruleoalba*. Inset: intranuclear strong c-fos positivity in a giant neuron of the VCN. Paraffin embedded brain sample. DAB counterstained with thionine. Atlantic spotted dolphin, *Stenella frontalis* (**a**) Left Hippocampus. Archicortex: DG, Gyrus dentatus and CA, cornu Ammonis. Cplex, choroid plexus, lv, lateral ventricle. Thionine. Atlantic spotted dolphin, *Stenella frontalis* (**b**); Hypothalamic tuberal area: supraoptic nucleus (SON) and suprachiasmatic (SC) nucleus. ot, optic tract. Inset: neurons of the SON. Free-floating, not embedded brain sample. DAB. Vasopressin. Blainville’s beaked whale, *Mesoplodon densirostris* (**c**); Substantia nigra and crus cerebri. Distribution of the Tyrosine Hydroxylase immunoreactive neurons (arrowheads), which make up the mesencephalic substantia nigra. crce, crus cerebri; cC, caudal colliculus. Free-floating, not embedded brain sample. DAB. Atlantic spotted dolphin, *Stenella frontalis* (**d**); nNOS immunoreaction in the cortical neurons. Paraffin embedded brain sample. DAB not counterstained. Striped dolphin, *Stenella coeruleoalba* (**e**) Parvalbumin immunoreaction in the cortical neurons. Paraffin embedded brain sample. DAB not counterstained. Striped dolphin, *Stenella coeruleoalba* (**f**) Alpha-synuclein immunopositive round body in the neuropil of the mesencephalon. Inset: Ubiquitin immunopositivity in the same round body. Paraffin embedded brain sample. AEC counterstained with Mayer’s hematoxylin. Short-finned pilot whale, *Globicephala macrorhynchus* (**g**) Purkinje cells show diffuse granular and vacuolar (arrowhead) cytoplasmic NFT-positive labeling. Free-floating immunolabeling. DAB counterstained with thionine. Blainville’s beaked whale, *Mesoplodon densirostris* (**h**) Vascular amyloid deposition in the amygdaloid complex. Free-floating immunolabeling. DAB counterstained with thionine. Atlantic spotted dolphin, *Stenella frontalis* (**i**); Mild intranuclear TUNEL positivity in some multipolar small spherical and multipolar globular neurons (arrowheads) of the VCN. Counterstained with thionine. Cuvier’s beaked whale, *Ziphius cavirostris.* Inset: Positive control, VCN incubated with TACS-Nuclease. Neuronal and glial nuclear positivity. Counterstained with thionine. Newborn bottlenose dolphin, *Tursiops truncatus*. (**j**); Ubiquitin nuclear staining (arrowheads) in the anterior part of the VCN. Paraffin embedded brain sample. DAB counterstained with thionine. Atlantic spotted dolphin, *Stenella frontalis* (**k**).

**Figure 5 vetsci-09-00038-f005:**
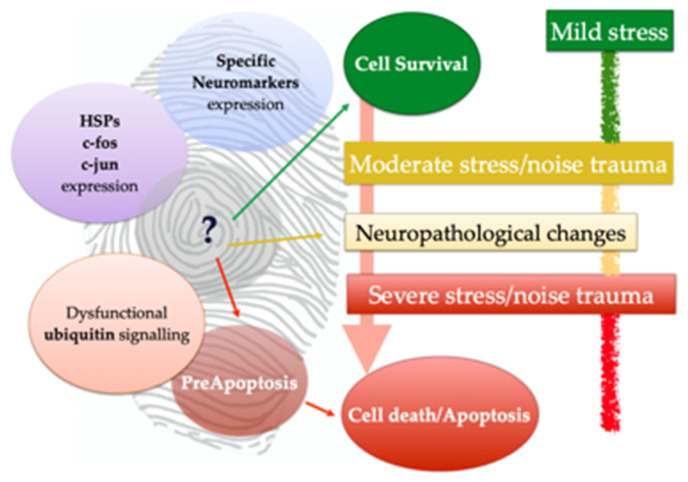
Schematic representation of fingerprint-like cellular identikit, of “healthy” vs. pathological neurons. Adapted from [69]. HSPs, Heat Shock Proteins.

**Table 1 vetsci-09-00038-t001:** The table shows 20 primary antibodies used as neuromarkers in the cetaceans’ brains both for ff-IHC and/or p-IHC. monoclonal (M); polyclonal (P); free-floating immunohistochemistry (ff-IHC); formalin-fixed paraffin-embedded samples (FFPE); formalin fixed cryosectioned samples (FFCS); Mouse (Mo) Goat (G); Rabbit (R); Horse (H); Not applicable (N/A); Normal Serum (NS); Secondary Antibody (SA); Biotinylated anti-mouse IgG (BaMo); Biotinylated anti-rabbit IgG (BaR); Biotinylated anti-goat IgG (BaG); minutes (min).

Primary Antibody	Type	Specificity [Published Manuscript]	Diluition ff-IHC p-IHC	Antigen Retrieval (Only for FFPE)	NS	SA
c-Fos (4) Santa Cruz Biotechnology sc-52	P/R	c-Fos Protein	1/100 (both)	Pronase, 7 min	G	BaR
c-Jun (D) Santa Cruz Biotechnology sc-44	P/R	c-Jun Protein	1/100 (both)	Citrate Buffer, 90–95 °C, 10 min (pH 6)	G	BaR
HSP70 Abcam Ab6535	M/Mo	Heat Shock Protein 70 kD [35]	1/100 (both)	None	H	BaMo
Ubiquitin Dako Z045801	P/R	Human Ubiquitin	1/100 (p-IHC)	None	G	BaR
Neuroglobin Abcam Ab37258	M/Mo	Neuroglobin	1/100 (p-IHC)	None	H	BaMo
Calretinin Swant 6B3	M/Mo	Calretinin calcium-binding protein [35]	1/500 (p-IHC)	Wet autoclave method of Shin 118° C, 5 min	H	BaMo
Calbindin D-28k Swant 300	M/Mo	Calbindin calcium-binding protein	1/500 (both)	Wet autoclave method of Shin, 118° C, 5 min	H	BaMo
Parvalbumin Swant 235	M/Mo	Parvalbumin calcium-binding protein	1/500 (p-IHC)	Wet autoclave method of Shin, 118° C, 5 min	H	BaMo
GFAP DakoCytomation	P/R	Glial Fibrillary Acidic Protein	1/120 (p-IHC)	None	G	BaR
nNOS Millipore Ab5380	P/R	Nitric Oxide Synthase	1/300 (p-IHC)	Wet autoclave method of Shin, 118° C, 5 min	G	BaR
TH Monosan MONX10786	M/Mo	Tyrosine Hydroxylase [14]	1/200 (ff-IHC) 1/50 (p-IHQ)	Wet autoclave method of Shin, 118° C, 5 min	H	BaMo
CRF Abcam Ab59023	P/G	Corticotropin Releasing Factor	1/100 (ff-IHC)	N/A	R	BaG
Vasopressin Abcam Ab39363	P/R	Vasopressin	1/500 (ff-IHC)	N/A	G	BaR
HSV1 Abcam Ab9533	P/R	Herpesvirus type I [29]	1/50 (p-IHC)	Pronase 10 min	G	BaR
CDV VMRD CDV-NP	M/Mo	Nucleoprotein of Canine Distemper Virus [36]	1/100 (p-IHC)	Wet autoclave method of Shin, 118° C, 5 min	R	BaMo (1/20)
Laforin Novus Biologicals NBP2-24474	P/R	Human Laforin (EPM2A)	1/100 (p-IHC)	Wet autoclave method of Shin, 118° C, 5 min	G	BaR
B-Amyloid Invitrogen 700254	M/R	Beta Amyloid (H31L21) [15]	1/100 (ff-IHC)	N/A	G	BaR
NFT AHB0161	P/R	Neurofibrillary Tangles [15]	1/100 (ff-IHC)	N/A	G	BaR
α-Synuclein Abcam Ab27766	M/Mo	Alpha-synuclein (LB 509)	1/100 (ff-IHC)	N/A	H	BaMo
α-Synuclein Invitrogen 35-8300	M/Mo	Alpha-synuclein (Syn 505)	1/100 (p-IHC)	Wet autoclave method of Shin, 118° C, 5 min	H	BaMo

## Data Availability

The data presented in this study are available from the corresponding author on reasonable request. Some data may be part of other unpublished studies.

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
