# Peer review of "Methodology and Neuromarkers for Cetaceans’ Brains"

_vetsci, 2022, doi:10.3390/vetsci9020038_

Round 1
Reviewer 1 Report
See attached file.

Reviewer 2 Report
I applaud the author’s efforts to standardize fixation of these larger cetacean brains. Methods of fixation may depend on the goals or means of best determining pathology. For example, post fixation MRI has become useful for pathology (Lemaitre below, and many papers on human postmortem brain MRI).
Line 33, reword. Cetacea spend all of their time in the water. In fact, lines 29 to 33 are so confusing they should be left out since the paper is about cetacean brains.
Line 40. In general the statement is wrong. Some cetaceans have large olfactory lobes, for example the bowhead whale. Even those that don’t have olfactory lobes, have a terminal nerve system.
Thewissen, J. G. M., George, J., Rosa, C., & Kishida, T. (2011). Olfaction and brain size in the bowhead whale (Balaena mysticetus). Marine Mammal Science, 27(2), 282-294.
Line 43. All odontocetes do not have large brains in absolute terms or relative to body size. There is a great deal of variation. See, Ridgway et al 2016 cited below.
Line 48. The largest brains reported in sperm whale and killer whales respectively are 9200 and 9300 grams. See the two citations below:
Jacobs MS, Jensen AV (1964): Gross aspects of the brain and a fiber analysis of cranial nerves in
the great whale. J Comp Neurol 123: 55–71. (sperm whale)
Shindo N (1975): History of Whales in the Inland Sea . Hyogo, Hyogo Dietists’ Institute. (killer whale)
Line 58. Do you have a reference for this statement?
Line 72. Reference 17, the paper by Roth and Dicke (2005) contains a lot of speculation and is not an original source. Most of their data come from Haug 1987. Without any original data, Roth and Dicke claim that information processing capacity of cetaceans is much lower because of a thin cortex, low neuron packing density and low axonal conduction velocity. In fact, real data on axonal conduction velocity shows just the opposite. See:
Popov, V. V., & Supin, A. Y. (1990). Auditory brain stem responses in characterization of dolphin hearing. Journal of Comparative Physiology A, 166(3), 385-393.
Reference 15 used a different fixative. “The brain was immersed fixed in 10l of Streck Tissue Fixative (Streck Laboratories Inc., Omaha, NB, USA). The whole brain, in the fixative, was placed on a floating pier for 2 months so that wave action produced a rocking motion to improve fixation of the large brain.”
Ridgway, S. H., Carlin, K. P., Van Alstyne, K. R., Hanson, A. C., & Tarpley, R. J. (2016). Comparison of dolphins' body and brain measurements with four other groups of cetaceans reveals great diversity. Brain, Behavior and Evolution, 88(3-4), 235-257.
[used Ringer’s solution as a more physiological formalin mixture and measured change in brain mass with time in formalin solution. They made phosphate-buffered 10% formalin with a specific gravity of 1.029. They suspended cetacean brains in this solution with a thread attached to the blood vessels at the base of the brain. Putting the immersed brain on a shaker or even an ocean pier allows gentle movement and better penetration of the fixative. ]
Cozzi, B., Mazzariol, S., Podestà, M., Zotti, A., & Huggenberger, S. (2016). An unparalleled sexual dimorphism of sperm whale encephalization. International Journal of Comparative Psychology, 29(1).
Ridgway, S. H., Brownson, R. H., Van Alstyne, K. R., & Hauser, R. A. (2019). Higher neuron densities in the cerebral cortex and larger cerebellums may limit dive times of delphinids compared to deep-diving toothed whales. PloS one, 14(12), e0226206.
[used NISSL and GFAP for neurons and glia assessment to Ziphius, Orcinus and other large cetacean brains]
Korzhevskii, D. E., Sukhorukova, E. G., Kirik, O. V., & Grigorev, I. P. (2015). Immunohistochemical demonstration of specific antigens in the human brain fixed in zinc-ethanol-formaldehyde. European journal of histochemistry: EJH, 59(3).
A special fix for human brains
Kuiken, T.; García-Hartmann, M. Proceedings of the first European Cetacean Society Workshop on Cetacean Pathology: 563 dissection techniques and tissue sampling, Leiden, the Netherland. 1991. 564
Is this an available published work? I cannot find it.
Lemaitre, F., Fahlman, A., Gardette, B., & Kohshi, K. (2009). Decompression sickness in breath-hold divers: a review. Journal of sports sciences, 27(14), 1519-1534.
Round 2
Reviewer 2 Report
The manuscript is much improved. Some editing by an English speaker would help,